# Epidemiology, Diagnostic Strategies, and Therapeutic Advances in Diffuse Midline Glioma

**DOI:** 10.3390/jcm12165261

**Published:** 2023-08-12

**Authors:** Gloria Miguel Llordes, Víctor Manuel Medina Pérez, Beatriz Curto Simón, Irene Castells-Yus, Silvia Vázquez Sufuentes, Alberto J. Schuhmacher

**Affiliations:** 1Molecular Oncology Group, Instituto de Investigación Sanitaria Aragón (IIS Aragón), 50009 Zaragoza, Spain; 2Pediatric Cancer Center Barcelona, Hospital Sant Joan de Déu, 08950 Barcelona, Spain; 3Hospital Universitario de Cruces, 48903 Barakaldo, Spain; 4Hospital Universitario Miguel Servet, 50009 Zaragoza, Spain; 5Fundación Aragonesa para la Investigación y el Desarrollo (ARAID), 50018 Zaragoza, Spain

**Keywords:** DMG, DIPG, diagnosis, treatment, molecular classification, histone 3, clinical trials

## Abstract

**Object:** Diffuse midline glioma (DMG) is a highly aggressive and lethal brain tumor predominantly affecting children and young adults. Previously known as diffuse intrinsic pontine glioma (DIPG) or grade IV brain stem glioma, DMG has recently been reclassified as “diffuse midline glioma” according to the WHO CNS5 nomenclature, expanding the DMG demographic. Limited therapeutic options result in a poor prognosis, despite advances in diagnosis and treatment. Radiotherapy has historically been the primary treatment modality to improve patient survival. **Methods:** This systematic literature review aims to comprehensively compile information on the diagnosis and treatment of DMG from 1 January 2012 to 31 July 2023. The review followed the PRISMA (Preferred Reporting Items for Systematic Reviews and Meta-Analyses) statement and utilized databases such as PubMed, Cochrane Library, and SciELO. **Results:** Currently, molecular classification of DMG plays an increasingly vital role in determining prognosis and treatment options. Emerging therapeutic avenues, including immunomodulatory agents, anti-GD2 CAR T-cell and anti-GD2 CAR-NK therapies, techniques to increase blood–brain barrier permeability, isocitrate dehydrogenase inhibitors, oncolytic and peptide vaccines, are being explored based on the tumor’s molecular composition. However, more clinical trials are required to establish solid guidelines for toxicity, dosage, and efficacy. **Conclusions:** The identification of the H3K27 genetic mutation has led to the reclassification of certain midline tumors, expanding the DMG demographic. The field of DMG research continues to evolve, with encouraging findings that underscore the importance of highly specific and tailored therapeutic strategies to achieve therapeutic success.

## 1. Introduction

Diffuse midline glioma (DMG) is a type of malignant tumor derived from glial cells, primarily occurring in children. It is characterized by a diffuse infiltrative growth of tissue affecting the brain stem, which often renders surgical resection unfeasible [1]. Brain stem tumors are relatively prevalent among children, accounting for approximately one in five cases of nervous system tumors in this age group. Of these, approximately four out of five are DMGs. The clinical symptoms in this type of tumor depend to a large extent on the brain areas affected producing sensory or motor syndromes such as facial paralysis, balance disorders, visual disturbances, dysarthria, dysphagia, and dysphagia, among others [2,3,4].

From a histological perspective, it is a highly heterogeneous disease with a wide range of histological variations. These tumors exhibit significant diffusion and frequently invade neighboring brain structures beyond the pons. Numerous studies have reported that leptomeningeal dissemination and subventricular spread are common findings in up to one-third of DIPG cases. In fact, tumor cells have been identified as far rostrally as the frontal lobe, highlighting the extensive nature of the disease [5]. Like other brain stem tumors, the degree of malignancy is determined by the differentiation level of the tumor cells, with low-grade tumors exhibiting higher differentiation and high-grade tumors characterized by poorly differentiated glial cells and significant anaplasia [5,6].

Previously and still commonly referred to as diffuse intrinsic pontine glioma (DIPG), this tumor was formerly recognized as a grade IV brain stem glioma and has recently been renamed according to the World Health Organization (WHO) CNS5 nomenclature as “diffuse midline glioma” [6].

Indeed, it is essential to emphasize the significant distinction in diagnosing DMG and DIPG. The publication of WHO CNS5 in 2021 has resulted in a noteworthy change. WHO grade IV tumors located in the thalami, brainstem, and spinal cord of both pediatric and young adult populations are now reclassified as DMG, particularly if they bear the H3-K27 mutation. Recent advancements in tissue acquisition and molecular profiling have revealed that DMG and GBM (glioblastoma multiforme) are distinct disease entities with separate tissue characteristics and genetic profiles. Additional profiling has shed light on the origin of the disease and the influence of various mutations, such as the highly recurring K27M mutation in histone H3, on its tumorigenesis. Furthermore, early evidence suggests that DMG has a unique immune microenvironment, characterized by low levels of immune cell infiltration, inflammation, and immunosuppression, which may have implications for the disease development and outcomes [7,8].

The reclassification of tumors into the DMG category not only impacts the patient demographic but also broadens the disease’s classification to encompass not only pediatric cases but also those affecting young adults [9].

In the past decade, the primary method for diagnosing DMG has been magnetic resonance imaging (MRI) [1,2,3,4,10]. The tumor grows within the brain stem, responsible for vital functions like breathing, heart rate, and blood pressure. Therefore, initially, the diagnosis relied solely on MRI since the distinct imaging characteristics of DMG provided definitive diagnosis. However, in recent years, biopsies have become increasingly important for histological confirmation and, more significantly, for molecular biology studies.

The combination of a high mortality rate and a lack of effective therapeutic options renders DMG one of the most challenging tumors in the field of pediatric neuro-oncology. By the early 2010s, the estimated survival rate for DMG treatment stood at a mere 10% two years after diagnosis [5,11].

In terms of effectiveness, the available therapies used for this type of tumor at the beginning of the previous decade were not effective enough in preventing tumor relapse and achieving long-term survival [2,3,4]. Achieving satisfactory treatment outcomes for DMG poses significant challenges, largely due to its location and high likelihood of developing refractory disease. In addition, its infiltrative and diffuse patterns further complicate surgical intervention, making resection impossible and diminishing the efficacy of radiotherapy [3,4,5,6]. Due to these challenges, stereotactic radiotherapy has become a common approach for treating DMG. However, its effectiveness is often limited due to the infiltrative nature of the tumor and the challenge of targeting all malignant cells [3].

Despite being classified as a rare tumor, DMG accounts for approximately 10–15% of all brain tumor deaths in children, representing one of the most important causes of tumor-related deaths in this age group [1,11,12], so is particularly relevant today due to the high mortality rate and the lack of effective therapeutic options. DMG has emerged as a compelling subject of scientific investigation, particularly in the realm of clinical studies and trials aimed at assessing novel therapeutic approaches such as immunotherapy, T-cell therapy, and targeted therapies. However, substantial further research is still required to improve the prognosis of patients with DMG [13,14].

The prognosis for DMG has remained discouraging for over five decades and biological research has been hindered by the scarcity of pre-treatment tissue samples, as biopsies were typically reserved for atypical cases [2,3,4,15]. However, recent advances in surgical techniques and the possibility of performing molecular analysis have improved the safety and potential utility of biopsies. Brainstem biopsies have now been incorporated into several prospective clinical trials [16]. These and other recent efforts have yielded new insights into the molecular pathogenesis of DMG and have opened up new avenues for research.

Ongoing clinical studies are currently investigating promising therapeutic approaches (Appendix A), such as T-cell therapy, immunotherapy, and targeted therapies [11,16]. Consequently, this comprehensive review of the existing literature on DMG aims to gather reliable information that can contribute to improving the prognosis of DMG patients in the long term. Despite the existence of other reviews that specifically address DIPG/DMG [17,18,19,20] (Appendix A), this review of DMG is necessary to provide updated insights, as this field has been rapidly evolving in recent years and the recent advances can potentially improve patients’ survival.

A comprehensive review study is proposed to address key aspects related to this tumor such as its definition, epidemiological aspects, diagnostic methods, classifications, and treatment options. By examining these fundamental aspects, the review aims to contribute to the existing body of knowledge and provide valuable insights into DMG. Such research will be valuable to clinicians, researchers, and other stakeholders in the field, ultimately driving progress in DMG management and patient outcomes.

## 2. Methods and Search Strategy

In order to achieve the proposed objective, a systematic literature review of relevant studies published between 1 January 2012 and 31 July 2023 was conducted following the PRISMA (Preferred Reporting Items for Systematic Reviews and Meta-Analyses) statement. This bibliographical search was conducted using the PubMed, Cochrane Library, and SciELO (through the LILACS) databases.

The search terms used were: “Diffuse intrinsic pontine glioma”, “DIPG”, “Diffuse midline glioma”, “DMG”, “epidemiology”, “diagnosis”, “molecular biology”, “SPECT”, “biopsy”, “stereotactic biopsy”, “classification”, “radiotherapy”, “radiation therapy”, “proton therapy”, “chemotherapy”, “histone”, “H3”, “H3K27”, “M3.H3K27”, “New therapies”, “peptide vaccines”, and “CNS WHO”.

To structure the search in the different engines, we used the Boolean operator “AND” to connect two different terms and “OR” was employed to represent an alternative between one term and another. Inclusion criteria included addressing any of the following topics: diagnosis, treatment of DMG, epidemiology, molecular classification, or WHO tumor classification. Studies were included in English or Spanish, available for full reading online. Three authors independently screened the titles and abstracts of articles identified in the search and selected those that met the inclusion criteria. Any disagreement between authors was resolved through consensus.

The final inclusion of articles was determined following an assessment of their evidence quality using Sackett’s levels of evidence. This review has not been registered.

## 3. Results

Following the identification and screening process depicted in Figure 1, a total of 126 articles were included in this review.

### 3.1. DMG Definition

DMG is a rare and aggressive brain tumor that primarily affects children and young adults. It predominantly arises in the brainstem and is characterized by a remarkably low median overall survival time, with over 50% of affected individuals surviving less than 12 months after diagnosis [13,17,18,19,21,22,23,24]. DMGs represent a heterogeneous group of tumors that exhibit distinct biological features compared to other high-grade brain tumors [25]. With the addition of other midline structures to DIPG, the diagnosis of DMG is broader. Due to their unpredictable location and growth pattern, complete surgical resection of DMG is typically unattainable, posing significant challenges in pediatric neuro-oncology [13,24].

Damodharan et al. (2022) show that, although DMG is one of the tumors with the bleakest prognosis at present, in recent years the molecular study of DMG has advanced, leading to new avenues of research [24]. This tumor has been related to oncogenesis that is produced from proto-oncogenes and the suppression of specific cell differentiation [17,18]. Advanced diagnostic strategies not only hold a crucial role in confirming the diagnosis of DMG but also possess significant potential in guiding personalized treatment approaches, ultimately improving patient outcomes.

### 3.2. Epidemiology

According to Argersinger et al. (2021) and other authors [22,23,24,25], DMG is recognized as one of the most malignant brain tumors in childhood, comprising over two-thirds of all brainstem tumor cases. It has been reported that DMG may contribute to up to 15% of all childhood brain tumor-related deaths [23]. The survival rate for DMG is particularly low, with only one in ten cases surviving beyond two years after diagnosis [22,23,24,25,26].

Approximately one in five childhood brain tumors is identified as a form of DMG and nearly four in five of all brainstem tumors fall into this category [17,23]. In the United States alone, 150 to 400 new cases are reported annually [23,24,26].

### 3.3. Diagnostic Methods

#### 3.3.1. Magnetic Resonance Imaging (MRI)

The conventional MRI used for diagnosing DMG typically reveals hypointense signals on T1-weighted images and hyperintense signals on T2-weighted images, without contrast enhancement (Figure 2). This characteristic distinguishes DMG from other brain stem tumors. The tumor frequently localizes in the brain stem pons, occupying more than two-thirds of the total volume of the pons and sometimes extending to surrounding areas [23,26]. Enhancement techniques for imaging vary based on the tumor type and have not been standardized for cases with specific mutations [23].

Some of the most common macroscopic findings on imaging techniques for DMG include necrosis in one in five cases, growth from the cephalocaudal direction towards the midbrain, cerebral peduncles, cerebellum or medulla oblongata, and poorly defined margins [10,26].

These tumors are usually exophytic, solid, and outgrowing, compressing the basilar artery, with rare involvement of the meninges [6,26]. Spectroscopy may show significantly higher choline/N-acetylaspartate and creatine/choline ratios compared to other tumors. Contrast MRI is used after radiotherapy, as radiation necrosis favors contrast uptake [19,26].

#### 3.3.2. Biopsy and Laboratory Diagnostic Techniques

Obtaining an accurate histological sample for DMG diagnosis poses a significant challenge, necessitating the use of a reliable and safe method such as stereotactic biopsy. This approach enables precise collection of tissue samples from the tumor, facilitating histological examination and molecular analysis (Figure 3), leading to improved diagnostic accuracy [22,23]. Additionally, the analysis of the tumor’s surrounding serum for the presence of tumor cells or specific biomarkers has emerged as a promising diagnostic technique [22,23].

Recent evidence suggests that biopsy is not only safe and diagnostically accurate, but also has the potential to influence treatment decisions for individual patients [27,28]. Advancements in neuroimaging techniques, like diffusion tensor imaging and frameless stereotactic technologies, have led to a re-evaluation of performing biopsies on brainstem tumors [29,30,31,32].

The safety of brainstem biopsy further supports the feasibility of obtaining biopsy-derived DIPG/DMG models for research purposes. Various approaches, including supratentorial transfrontal, infratentorial, and transcerebellar–transpeduncular, have been applied with high diagnostic accuracy and low morbidity in brainstem lesions [29,30,31,32]. The selection of a specific trajectory according to the lesion’s location is crucial when approaching brainstem lesions.

The majority of models used for studying DIPG and diffuse midline glioma, particularly those with the H3 K27M mutation (DMG), are either derived from autopsies or genetically engineered [31,32,33,34]. However, these models have limitations for translational studies. To address these limitations, utilizing biopsy tissue for developing laboratory models offers a unique opportunity to create systems that mimic tumors in their untreated state, thereby replicating the initial disease condition.

The advancements in analyzing DMG tumor tissue have provided researchers with a deeper understanding of the disease biology, which may lead to the development of more effective treatment strategies. Stereotactic biopsy, despite yielding limited tumor tissue due to the size of the needle, has demonstrated its ability to provide sufficient material for histopathologic diagnosis and immunohistochemical staining [30,31,32]. As such, performing a biopsy in any lesion suspected of DIPG is highly recommendable, as the new molecular classification and therapeutics under investigation may offer an opportunity for the patient to be included in ongoing clinical trials.

**Figure 3 jcm-12-05261-f003:**
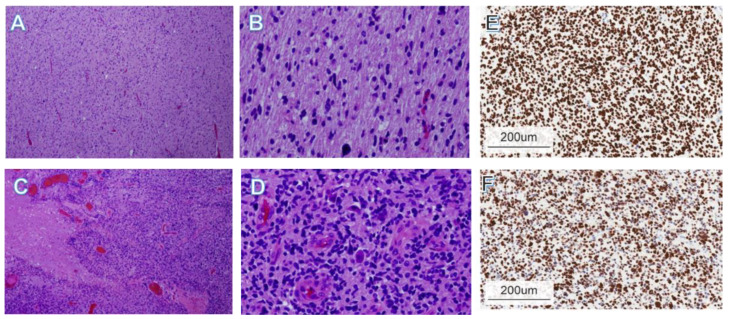
Representative histology of pediatric DMG. Histologic geographic variability of DMG. ×4 (**A**,**C**) and ×20 (**B**,**D**). Hematoxylin and eosin stains from different sections of a single tumor showing low-grade (**A**,**B**) and high-grade (**C**,**D**) areas. (**E**) Immunohistochemical staining of the same sample using antibody directed to the H3K27M epitope. The sample is strongly positive. (**F**) Immunohistochemical staining of mutant p53, a common co-occurring mutation in DMG. (**A**–**D**) Reprinted with permission and adapted from Warren et al. (2012) [15] and (**E**,**F**) reprinted with permission and adapted from Srikanthan et al. (2021) [35].

#### 3.3.3. Molecular Studies

The advent of molecular studies has ushered in a new era in the diagnosis and management of DMG, as affirmed by multiple experts in the field. Molecular testing has emerged as a pivotal paraclinical tool, augmenting the diagnostic and prognostic capabilities of biopsy. At the forefront of its importance is the detection of H3K27M mutations in tumor DNA, which holds immense clinical significance and serves as a robust biomarker for DMG [18,19] (Figure 3E).

The determination of the H3K27M mutation through immunohistochemistry (IHC) is essential for enhancing the histological classification and prognosis of pediatric brain tumors. Initially, H3K27M mutations were exclusively identified in diffuse midline gliomas (DMGs), establishing a unique molecular hallmark for this specific condition. Subsequently, these mutations were also detected in other types of brain tumors. However, it seems that the clinical prognostic value associated with these mutations is particularly notable within the domain of diffuse midline gliomas [36,37].

The assessment of H3K27M mutation status extends beyond immunohistochemistry (IHC) and encompasses alternative techniques such as Sanger sequencing, next-generation sequencing (NGS), droplet–digital polymerase chain reaction, and pyrosequencing. Notably, diffuse midline gliomas (DMG) have been found to potentially harbor subclonal H3K27M mutations with mosaic patterns. Instances have arisen where tumor cells exhibited positivity in the cytoplasm or immuno-positivity in lymphocytes, only to be subsequently verified as H3 wild-type through Sanger sequencing. Consequently, there are situations wherein additional sequencing methods are imperative to accurately ascertain the H3K27M mutation status [38].

Genome-wide DNA methylation profiling has emerged as a crucial tool for diagnosing central nervous system (CNS) tumors. This innovative technique holds significant value for both pediatric and adult patients. By classifying brain tumors according to their unique DNA methylation patterns, this approach has revolutionized how we diagnose cases where traditional histology alone may not provide clear answers. This advancement has notably transformed the diagnostic strategy for CNS tumors with uncertain characteristics [37,39,40,41].

The 2021 WHO Classification of Tumors of the Central Nervous System incorporates various tumor types and subtypes that require, in part, the use of whole genome methylation profiling for accurate diagnosis. The contemporary methodology for array DNA methylation profiling involves employing a reference library of tumor DNA methylation data, coupled with a machine learning-based tumor classifier. This pioneering approach was introduced and popularized by the German Cancer Research Network (DKFZ) and University Hospital Heidelberg. Remarkably, this research collective has generously made their CNS tumor classifier accessible for research purposes through an online platform [42,43].

Furthermore, the application of high-density methylation arrays has emerged as a robust technique for effectively categorizing primary brain neoplasms into distinct molecular subgroups. The development of the DKFZ/Heidelberg CNS tumor methylation classifier significantly contributes to our comprehension of methylation patterns and their interplay with clinical variables. This advancement greatly enhances the precision of molecular pathological classification in the realm of brain tumors [42,44].

A study conducted by Ceccarelli et al. (2016) [45] on diffuse gliomas revealed that these epigenetic subgroups offer prognostic insights independent of factors like age and tumor grade. This underscores the significant value of epigenetic information in predicting disease outcomes and tailoring treatment strategies; they used the Illumina Infinium Human Methylation array platform. The first version of this platform 450 (450 K) array not only enables methylation profiling but also facilitates concurrent copy number analysis. Additionally, a more recent validation has been carried out with the 850 K array. This application has led to the identification of six distinct subgroups, each enriched for specific DNA mutations. Notably, these subgroups exhibit discernible clinical characteristics encompassing factors such as age, anatomical location, and treatment outcomes [45].

### 3.4. Previous and Current WHO Classification of DMG

Until 2016, all DMGs were classified solely based on histological and morphological MRI characteristics of the tumor [25,46,47]. At the time, tumor grading could be summarized as: low grade or grade I and grade II tumors were those with low proliferative potential. Whereas grade I tumors could be possibly cured via surgical resection, grade II were usually infiltrative and tended to recur and progress to higher grades. Grade III tumors were those considered malignant, determined by histological evidence of mitosis or anaplasia, with frequent recurrence and rarely curable. High grade or grade IV tumors were those malignant and highly proliferative with active mitosis, and prone to necrosis, which is associated with a rapid evolution and therefore fatal prognosis causing death within months [6,48,49].

In most cases, DMGs were considered high grade due to their histological appearance (such as glioblastoma multiforme grade IV or anaplastic astrocytoma grade III), but this classification was incorrect, as tumors such as well-differentiated astrocytoma (grade II) were considered to be lower grade lesions.

Recognizing the limitations of this approach, a paradigm shift occurred in 2016 when molecular classifications emerged from earlier discoveries, prompting a re-evaluation of DMG severity based on both histological and molecular–genetic characteristics [5,6,17,22,50,51]. The 2016 CNS WHO classification considers phenotypic and genotypic parameters. This additional layer of refinement results in narrowly defined tumor types, creating larger groups of entities that do not fit in any category, designated NOS (not otherwise specified). The combination of both histology and genetics creates the possibility of discordant results. Therefore, the 2016 classification still considers not possible to understand nosological and clinical significance of certain genetic changes without histology. The genotype is integrated with the phenotype but still maintaining histology as the main criteria for grade determinations, as well as for the understanding of NOS tumors [49].

The periodicity of major WHO classification updates depends on many factors. However, the acceleration in our understanding of the molecular features of CNS tumors and their clinical impact has created a need for more frequent updates in the past years. Intended to fill the gap between official WHO classifications, the cIMPACT-NOW (Consortium to Inform Molecular and Practical Approaches to CNS Tumor Taxonomy—Not Official WHO) was created, in order to provide possible guidelines for clinicians and waypoints for the next WHO classification update [48]. Since the publication of the 2016 CNS WHO update, seven cIMPACT-NOW updates have reviewed newly acquired criteria on CNS tumors classification, including the definition and diagnostic pathways of NOS tumors, and the relevance of the mutations in histone 3 (K23M mutation), B-Raf (V600E mutation), and isocitrate dehydrogenase (IDH), among others [52].

These updates are reflected in the revised classification scheme by the WHO published in 2021, which now integrates the histological and immunohistochemical features of DMG mutations [24]. Notably, recent molecular biology studies have led to the identification of histone 3 or “H3” mutants as a distinct subgroup within DMG. Among the various mutations studied, H3K27M has gained prominence and has been incorporated into the WHO classification of central nervous system tumors as a sign of severity [6,21,22,47,51].

There are three types of genetic modifications in histone 3 mutated DMG, which are summarized in the Table 1 with their respective frequencies.

H3K27M mutations in H3.3/H3.1 have been extensively documented by multiple authors, with a prevalence of ~80% in cases of DMGs, making it the most frequently observed mutation [22,23,50]. H3K27M represents a recurrent somatic gain-of-function mutation characterized by the substitution of lysine 27 to methionine (p.Lys27Met: K27M) within the histone 3 (H3) protein. Consequently, the classification of these tumors is commonly referred to as H3 classification [5,6,51,54]. Additionally, ACVR1 mutations have been identified in approximately 25% of DIPGs, contributing to the genetic landscape of these tumors [22,23,50].

In addition to H3K27M mutations, the new classification includes several other high-grade tumors. These include diffuse hemispheric glioma with H3G34 mutation, pediatric-type diffuse high-grade glioma with H3-wildtype and IDH-wildtype, and infantile-type hemispheric glioma [46]. Tumors lacking the H3K27M mutation are referred to as “wild-type” DMGs and they generally have a more favorable prognosis compared to those with the mutation, as the H3 mutation is associated with higher mortality [22,50,54]. In addition to a different biological behaviour caused by this mutation, the worse prognosis could be partially attributed by the midline localization of H3K27 tumors, which induces severe symptoms and prevents effective treatments as resection.

The TP53 mutation is frequently observed in wild-type DMGs, as highlighted by Buczkowicz and Hawkins in 2015 [5]. Another important mutation is ACVR1, which is considered a major factor in tumorigenesis and is present in nearly one-third of DMGs with the H3K27M mutation [22]. Tumors with the ACVR1 mutation may exhibit encoding of serine/threonine kinase (ALK2), while mutations in EZH2, an enzyme involved in histone methylation, have been associated with a very poor prognosis.

### 3.5. Treatments

#### 3.5.1. Radiotherapy (RT)

Radiotherapy has been the mainstay of treatment for improving the life expectancy of patients with DMG [55,56,57]. The standard guideline recommends delivering doses of 54 to 60 Gy over a period of 6 weeks upon DMG diagnosis. This approach, in practice for over 20 years, has been observed to halt tumor progression for approximately 3 months in nearly four out of five cases [13,21,24,25,47,58,59,60,61]. However, DMGs are highly aggressive tumors and, even with early initiation of RT, overall survival is minimally affected, as noted by Pai Panandiker et al. (2014) [61].

Hyperfractionated RT has shown potential in improving clinical management by caregivers and has a significant positive impact on the quality of life for patients with DMG [8,15]. In contrast, a study by Park et al. (2020) compared hyperfractionated therapy with standard RT and found similar survival outcomes for patients with DMG [62].

A phase III non-inferiority randomized trial comparing hypofractionated (39 Gy in 13 fractions and 45 Gy in 15 fractions) and conventional fractionated (54 Gy in 30 fractions) radiotherapy was performed [63]. They conducted a randomized trial involving 253 patients, divided into three arms receiving different radiation regimens. The findings revealed median overall survival (OS) rates of 9.6, 8.2, and 8.7 months in the groups receiving 39 Gy in 13 fractions, 45 Gy in 15 fractions, and conventional radiation, respectively. The study met their non-inferiority assumption (with a non-inferiority margin of 15%) at 18 months OS. After these results, hypofractionation radiotherapy can be considered for patients with DMG, with a dose of 39 Gy in 13 fractions, at the clinician’s discretion, taking into account the patient’s condition and tumor location.

#### 3.5.2. Proton Therapy

Proton therapy is a form of particle radiotherapy that offers precise dose localization and reduced late toxicity by minimizing radiation exposure to normal brain tissues surrounding the tumor. This treatment modality allows for safe irradiation of tumors that cannot be effectively treated with photon radiotherapy, as critical organs can be spared due to its high degree of dose conformity [64]. In cases of reirradiation, proton therapy can mitigate the risk of radiation necrosis in the surrounding brain tissue [64,65].

Currently, proton therapy is typically administered using a similar schedule to photon radiotherapy. Early studies with limited patient follow-up have demonstrated comparable therapeutic efficacy between proton therapy and photon radiotherapy. Mizumoto et al. (2016, 2017) conducted studies in Japan on pediatric malignancies treated with proton therapy and reported a low incidence of late toxicity [66,67].

However, there is a lack of studies with sufficient sample sizes and long-term follow-up to compare the long-term effects of proton therapy and photon radiotherapy. Therefore, in-depth studies involving large cohorts of children and extended follow-up periods are needed to fully assess the potential benefits of proton therapy in reducing long-term side effects.

#### 3.5.3. Chemotherapy (CTX)

DMG is distinguished by the presence of diverse tumor tissue, often accompanied by an intact BBB, which might account for its resistance to chemotherapy [9]. CTX is commonly administered in conjunction with RT to leverage their synergistic effect in treating DMG. To date, there are still no guidelines for both chemo- and radiotherapy, resulting in a wide variety of treatment schedules, particularly when facing disease progression.

Some evidence supports the combination of treatment using radiosensitizers like gemcitabine and capecitabine, originated from the results of two phase II trials [68,69], although their impact on overall mortality is minimal or negligible.

Kilburn et al. (2018) conducted a phase II trial with capecitabine, an alkylating agent, given alongside radiotherapy, followed by adjuvant capecitabine in 44 patients with newly diagnosed DIPG [68]. The findings showed earlier progression, with a 1-year progression-free survival (PFS) of 7.21% (SE = 3.47%) in the capecitabine-treated group compared to 15.59% (SE = 3.05%) in the historical control group (*p* = 0.007). While there was no significant survival benefit (overall survival (OS) and PFS were comparable to historical controls), the treatment regimen was well-tolerated. Similarly, Veldhuijzen van Zanten et al. (2017) conducted a phase I/II trial to evaluate the efficacy, safety, and tolerability of gemcitabine, a nucleoside metabolic inhibitor, during radiotherapy in nine children with newly diagnosed DIPG [69]. All patients experienced a reduction in tumor-related symptoms and quality of life (QoL) tended to improve during treatment. The PFS and median OS were 4.8 months (95% CI 4.0–5.7) and 8.7 months (95% CI 7.0–10.4), respectively. However, there were no significant survival benefits observed, with a median OS of 12.4 months and 8.7 months in intermediate- and high-risk patients, respectively.

Bailey et al. (2013) evaluated the efficacy of combining temozolomide (TMZ) with standard radiotherapy in children with DMG [59]. They administered concomitant temozolomide (75 mg/m^2^) during radiotherapy, followed by adjuvant temozolomide (75–100 mg/m^2^) for up to 12 cycles in a 21-day schedule. Their study did not demonstrate a survival improvement [59], which is consistent with findings reported by other authors [5,58,70]. However, Jackaki et al. (2016) tested the combination of oral TMZ with lomustine and observed significantly improved outcomes compared to TMZ alone [70].

Re-irradiation in combination with CTX or immunotherapy has been investigated [22,25]. These studies have shown a prolongation in overall survival compared to non-irradiated patients, although the observed changes, while statistically significant, have been modest. The data suggest that combining re-irradiation with other therapies may be more beneficial than implementing re-irradiation alone [22,25,65]. Other agents with no effect on overall mortality are described in the literature, such as veliparib, tested by Baxter et al. (2020) in a phase I/II study [60].

In addition to the aforementioned agents, other drugs frequently used in the treatment of DMG patients aim to alleviate symptoms or slow disease progression. These include etoposide, bevacizumab, irinotecan, nimotuzumab, and valproic acid. However, in most cases, these drugs have proven ineffective in significantly altering the fatal prognosis within a short period of time [13,58].

#### 3.5.4. Novel Approaches for Local Administration of Therapies

Recent advancements in technology and surgical techniques have enabled the exploration of novel approaches for local administration of therapies in the treatment of DMG. One such technique is convection-enhanced delivery (CED). Gwak and Park, 2017 and other authors have extensively discussed the potential of CED in DMG treatment [22,50,58,71], while preclinical trials using murine models have demonstrated the efficacy of CED in prolonging drug retention of multiple molecules at the infusion site and maintaining a sustained therapeutic effect against DMG [71]. However, it is important to note that there are currently no established guidelines regarding the therapeutic safety, potential systemic risks, or toxicity of this route of administration. Consequently, ongoing research and active debates in the field continue to explore these aspects [22,58,71].

Another recent therapeutic strategy that has gained attention is the utilization of magnetic resonance-guided focused ultrasound (MRgFUS) [72,73]. This innovative approach enables the targeted delivery of substances to the tumor site through the application of ultrasound waves in combination with microbubbles. By temporarily increasing the permeability of the blood vessels and the BBB, MRgFUS allows for localized delivery of therapeutic agents, making it a promising therapeutic option for the future [74,75].

MRgFUS holds significant potential in providing precise and non-invasive treatment for diffuse midline gliomas (DMGs). The ability to precisely target the tumor area and control the release of therapeutic substances offers advantages in terms of reducing systemic toxicity and enhancing treatment efficacy [74,76,77].

While MRgFUS shows promise, further research and clinical investigations are needed to optimize this approach, and evaluate its safety and effectiveness in DMG treatment. As this technique continues to evolve, it has the potential to revolutionize the field of neuro-oncology by providing a targeted and minimally invasive treatment option for patients with DMG.

#### 3.5.5. Targeted Therapies Based on the Molecular Classification

Drugs targeting deregulated histone acetylation and methylation (H3 mutations), such as GSKJ4, an H3K27M demethylase inhibitor that increases H3K27me3 in H3K27me3-expressing cells, have been tested in humans. These therapies aim to induce tumor cell death, arrest the cell cycle, modulate chromatin expression, and limit differentiation. Preclinical studies have shown improved outcomes when these therapies are combined with panobinostat, a histone deacetylase inhibitor [21,22,58,78]. A summary of preclinical studies can be found in Appendix A.

Large international collaborative efforts have facilitated the molecular analysis of a significant number of these rare tumors, revealing that H3K27M gliomas also encompass biologically and genetically diverse tumors. These comprehensive studies have led to the discovery of additional oncogenic driver alterations in H3K27M gliomas [79,80,81]. Interestingly, these events include mutations in well-described oncogenic pathways, such as cell-/DNA-damage repair mechanisms (TP53, PPM1D, ATM, and ATRX), and receptor tyrosine kinase signaling pathways (ACVR1, FGFR1, PIK3CA, PIK3R1, and BRAF). Double mutant gliomas with H3K27M and BRAFV600E alterations have been reported in numerous studies. This suggests a biological overlap between histologically defined low- and high-grade gliomas, and may be associated with a better prognosis than expected when compared to BRAF wildtype and H3K27-mutant DMGs [82,83,84,85,86,87].

On 16 March 2023, the FDA approved dabrafenib (Tafinlar, Novartis) in combination with trametinib (Mekinist, Novartis) for pediatric patients aged 1 year and older with low-grade glioma (LGG) harboring a BRAF V600E mutation and requiring systemic therapy (NCT02684058). The FDA also approved new oral formulations of both drugs suitable for patients who cannot swallow pills [88]. This marks the first FDA approval of a systemic therapy for the first-line treatment of pediatric patients with LGG carrying a BRAF V600E mutation. Patients with BRAF mutations may benefit from this therapy even if their tumors are not low-grade gliomas.

In tumors with ACVR1 mutations, ALK2 inhibitors have been identified as potential therapeutic agents, even when this mutation occurs in wild-type DMG. While in vitro studies have shown promise, there is a lack of in vivo studies in this context [22]. Similarly, EZH2 inhibitors and metabolic inhibitors such as the polyamide synthesis inhibitor difluoromethylornithine (DFMO) have shown efficacy in mouse models, but clinical studies are yet to be conducted [22,89,90]. Wei et al. (2018) conducted a comprehensive bioinformatics analysis on tumor composition leading to propose potential avenues for future therapies, highlighting the calcium signaling pathways and the neuroactive ligand-receptor interactions [91].

Additionally, the use of a dopamine receptor D2 (DRD2) antagonist and a caseinolytic protease P agonist (ClpP), known as ONC201, has been proposed as a potentially safe treatment option [92]. This specific DRD2/3 antagonist has the ability to penetrate the blood–brain barrier and demonstrates p53-independent effectiveness in preclinical models of high-grade glioma [93]. The mode of action of ONC201 involves triggering the integrated stress response while simultaneously deactivating Akt/ERK and other prosurvival signaling pathways [94]. Furthermore, ONC201 is capable of targeting and depleting cancer stem cells present in gliomas and other advanced cancers [95]. The first pediatric clinical trial of ONC201 confirmed that the adult recommended phase II dose, administered weekly through oral capsules and scaled by body weight, was well tolerated in pediatric patients. Results from this study indicate a safety and pharmacokinetic profile of ONC201 similar to that observed in adults. There was no dose-limiting toxicity, with most adverse effects being grade 1 or 2. Five patients (22.7%), all of whom started ONC201 after radiation and before recurrence, were alive at 2 years from the time of diagnosis [96]. New strategies are being studied with this drug as monotherapy and in combination with other agents such as PI3K inhibitors. While its use has been primarily after radiation or in cases of recurrence, the new strategies aim to incorporate ONC201 in earlier stages of the disease [97,98,99,100]. ONC201, panobinostat, and paxalisib are enzyme inhibitors that exhibit the potential to halt the growth of tumor cells by obstructing crucial enzymes required for cell proliferation. Currently, a phase II trial (NCI-2021-08386) is underway to evaluate various combinations of these drugs for the treatment of DMGs (NCI-2021-08386). Participants in the study will be randomly assigned to one of three study arms upon entry. Subsequently, they will be categorized into one to three phases and one of three cohorts based on their disease stage and prior treatment history. The target enrollment for the trial is 324 patients, with the estimated completion date set for 2027.

Another therapeutic strategy that has emerged in the field of neuro-oncology is the use of antibody–drug conjugates (ADCs) [101]. ADCs are a class of immunoconjugates that chemically link protein-specific antibodies with cytotoxic agents to selectively target cells expressing the antigen of interest with high specificity. However, the role of ADCs in patients with gliomas has been reconsidered due to various factors, such as toxicity associated with targeting the EGFR family with certain cytotoxic payloads and the challenge of penetrating the blood–brain barrier to reach tumor cells [101,102].

Several ADCs have been evaluated in clinical trials for gliomas. For example, depatuxizumab mafodotin (ABT-414), an ADC targeting EGFR, has shown potential in the treatment of recurrent gliomas [103]. However, a phase 3 trial did not demonstrate a benefit in overall survival when treating de novo glioblastomas with EGFR amplification [104].

Furthermore, nanobodies can be engineered to deliver therapeutic agents directly to glioma cells, enhancing the specificity and effectiveness of treatment while minimizing off-target effects. This targeted approach holds great promise for improving the outcomes of glioma patients and overcoming the limitations of conventional therapies.

The development of nanobodies targeting molecular targets in gliomas in addition to BBB shuttle peptides [105,106] represents an exciting advancement in neuro-oncology research and opens new possibilities for personalized medicine in the treatment of gliomas.

#### 3.5.6. Tumor Treating Fields

TTF utilizes low-intensity electric fields to inhibit cell division, showing efficacy in neoplastic conditions such as glioblastoma, pancreatic cancer, ovarian cancer, non-small cell lung carcinoma, and mesothelioma [107]. TTFs have been approved by the FDA as adjuvant therapy for newly diagnosed and recurrent glioblastoma [108,109]. Ongoing clinical trials are further exploring its potential in various malignancies, while interdisciplinary healthcare teams play a crucial role in the management of patients receiving TTF therapy.

An inherent limitation in the context of generating transcranial fields is the restricted depth of targeted tissue and the insulating impact of the skull [110,111]. An study by Ibn Essayed et al. (2023) investigates the potential use of TTFs for DMG focusing on the anatomical constraints associated with endoscopic endonasal implantation of an electrode array in the clivus region [112]. The findings reveal that, in the pediatric population studied, the dimensions allow for the placement of a 2.5 × 1 cm electrode array in 94% of patients (16/17), highlighting the feasibility of this approach. These results suggest that implantable transclival devices for TTF could serve as a viable adjunctive therapy for DMG, emphasizing the need for further investigation and development in this area.

#### 3.5.7. Immunotherapies

##### Vaccines

The development and utilization of autologous dendritic cell response-based vaccines offer a potential avenue for personalized immunotherapy in targeting specific tumor mutations and enhancing the immune response against cancer cells. Further research is warranted to explore the efficacy and long-term effects of this approach in clinical settings. An approach using peptide vaccines for this matter has been proposed and shown to be safe and to generate a specific immune response in the treatment of H3K27M mutated tumors [22]. An example of these approaches are peptide vaccines derived from neoantigens that occur within tumors. Mueller et al. (2020) demonstrated in a phase I/II clinical trial (NCT02960230) the efficacy of an H3.3K27M-targeted peptide vaccine, including newly diagnosed patients (3–21 years old) with HLA-A*02.01^+^ and H3-3K27M^+^ genotype [113]. The vaccine was well tolerated by patients and increased the overall survival of patients that showed specific anti-H3.3K27M^+^ CD8+ response. Another phase I clinical trial (NCT04749641) is currently in an enacting status using an H3.3K27M neoantigen vaccine.

Presently, there are other clinical trials that are being conducted for the study of other peptide vaccines for patients with DMG tumors, such as PEP-CMV (NCT05096481), composed of the synthetic long peptide Component A from human pp65, or rHSC-DIPGVax (NCT04943848), a mixture of peptides of neo-epitopes from the neo-antigen heat shock protein.

Virus-based vaccines are also under study, including DNX-2401 (Delta-24-RGD), an oncolytic adenovirus that has been genetically modified to induce in the patient a response similar to the natural immunity of a tumor, particularly in cases of H3K27M-mutated tumors [22,114]. Gállego Pérez-Larraya et al. (2022) conducted a single-centre dose escalation study in patients with newly diagnosed DMG. The study included 12 patients and two treatment regimens were employed: DNX-2401 viral particles: 1 × 10^10^ (for the first 4 patients) or 5 × 10^10^ (for the next 8 patients), with 11 receiving subsequent radiotherapy. The viral infusion was administered as a single dose through a catheter placed in the cerebellar peduncle. While the study demonstrated the feasibility of applying this therapy in humans, the authors emphasized the need for further research to establish safety profiles and assess potential adverse effects [114].

Despite the promising potential of oncolytic virotherapy, there remain several significant barriers that need to be addressed to enhance its efficacy. These obstacles encompass various factors, including viral tropism, delivery platforms, viral distribution, dosing strategies, antiviral immunity, and the oncolysis capability.

##### Immunomodulators

The use of some antibodies, such as nimotuzumab, administered simultaneously with standard-dose radiotherapy may have effects similar and comparable to those of radiotherapy plus chemotherapy. Kebudi et al. (2019) conducted a trial where they administered nimotuzumab alone in 17 cases and nimotuzumab combined with vinorelbine until death in patients with progressive disease. The survival rates were similar to those reported with standard treatment reported by other authors who describe treatment only with RT + CTX. Although the study did not serve to demonstrate superior outcomes, it established a basis supporting nimotuzumab as a well-tolerated drug without significant adverse effects [115]. Immunomodulatory antibodies such as MDV9300 (pidilizumab) have been used with promising results [22,116].

##### Immunotherapy with Tumor Infiltrating Lymphocytes (TILs)

Immunotherapy using tumor-infiltrating lymphocytes (TILs) has been considered as a potential treatment option for H3K27M-mutated tumors. TIL therapy involves extracting immune cells, particularly T cells, from the tumor tissue, expanding them in the laboratory, and then reinfusing them back into the patient with the aim of boosting the anti-tumor immune response [117,118].

However, the application of TIL therapy in H3K27M-mutated tumors has been limited due to challenges associated with obtaining a sufficient volume of tumor tissue through biopsy [118,119]. Collecting an adequate number of TILs is crucial for the success of this therapy, as it requires a significant quantity of tumor-infiltrating immune cells to be isolated and expanded. Given the diffuse nature of H3K27M-mutated tumors and their often limited accessibility, obtaining a suitable amount of tumor tissue for TIL extraction may be difficult [6,118].

##### CAR T-Cells and CAR-NK Anti GD2

Immunotherapy has gained ground in the area of nervous system cancer treatment research at least in the last 5 years. The molecular classification of these tumors has been very important in incorporating this knowledge in the development of chimeric antigen receptor (CAR) T-cell and NK therapies. Among them, CARs directed against the disialoganglioside GD2, which is an antigen that is almost uniquely associated with DMG tumor-related tissues, are showing some promise [22,50,120].

In vitro studies have demonstrated a significant reduction in the number of cells expressing the H3K27M mutation using GD2-targeting CAR T-cells [25,120]. Mount et al. (2018) conducted a study using orthotopic xenograft models of H3-K27M+ tumors and demonstrated that systemic administration of GD2-targeted CAR T-cells resulted in the elimination of grafted tumors, with only a small number of GD2-positive cells remaining [120]. Furthermore, Liu et al. (2022) recently described how the induction of immunogenic cell death in tumors is accompanied by the expression of molecular patterns associated with tissue damage in the tumor [121].

The possible penetration of the therapy into the brain tissue or the neurotoxicity of the therapy is uncertain, which has made it difficult to carry out human studies. The potential of immunotherapy as an adjuvant to traditional chemo-radiation has been proposed [22]. Majzner et al. (2022) conducted a well-designed phase I study testing GD2-CAR T-cell therapy for diffuse midline gliomas (DMGs) carrying the H3K27M mutation, which provided valuable insights into the therapeutic approach. Unlike other studies that focused on treatment-related inflammation, this study evidenced the development of signs and symptoms consistent with CAR T-cell-mediated inflammation (TIAN). The authors associated this inflammation with treatment toxicity, intracranial pressure alterations, and primary brain dysfunction [122].

The study findings confirmed the potential safety of GD2-CAR T-cell therapy as a therapeutic strategy for DMG. However, it is important to note that the experience is still limited and based on a small number of patients. Considering this, approximately 75% of the patients showed positive clinical and radiographic changes after the specific therapy improving the functionality of the infiltrated brain areas, a finding that was amplified in second therapies with GD2-CAR T-cells. As of the date of this review, a clinical trial in patients with H3K27M+ DMG and spinal DMG using GD2-CAR T-cell therapy is ongoing to determine the optimal dose, route, and schedule, and to determine the efficacy of GD2-CAR T-cell therapy in patients with H3K27M + DMG and spinal DMG [122].

## 4. Scope and Future Directions

In addition to all exposed above, the development of new tools to identify circulating biomarkers is on the rise. The quantification of tumor DNA in plasma has revolutionized treatment response monitoring, offering an accurate and non-invasive method for assessing therapeutic efficacy [22], facilitating the detection of such mutations.

The identification of molecular targets in gliomas has also sparked the development of nanobodies, the single variable domain-heavy chain fragment (VHH) of the heavy-chain-only antibodies (HCAbs) derived from the *Camelidae* species [92,123]. Nanobodies are designed to specifically bind to these molecular targets, such as overexpressed proteins or receptors in glioma cells. They offer several advantages over traditional antibodies, including their small size, high stability, and ability to penetrate the blood–brain barrier (BBB) [123,124].

Moreover, by targeting specific molecular markers in gliomas, nanobodies are emerging as a new tool with the potential to be used for various applications, including diagnostic imaging and targeted therapy. In diagnostic imaging, nanobodies can be labeled with imaging agents to selectively detect and visualize glioma cells using non-invasive imaging techniques including positron emission tomography (PET) or magnetic resonance imaging (MRI) [92,125,126].

This review highlights the importance of molecular classification in guiding therapeutic strategies for DMGs (Figure 4). Recent advances in understanding the genetic and molecular features of these tumors are leading to the development of targeted therapies, including immunotherapies and oncolytic vaccines, which show promising results in preclinical and early clinical studies.

However, there are limitations and challenges that need to be addressed. The infiltrative nature of DMGs, the limited accessibility to tumor tissue, and uncertainties regarding the penetration of therapies into the brain tissue pose obstacles to the implementation of effective treatments. Furthermore, the low frequency of DMGs and the limited number of patients available for clinical trials contribute to the slow progress in developing optimal therapeutic approaches.

Despite these limitations, ongoing clinical trials evaluating novel therapies based on the molecular characteristics of DMGs provide hope for improved outcomes. It is of importance to consider the geographic locations of clinical trials for potential treatment options. Some patients may be willing to travel for treatment and knowing the available trial locations can assist physicians in guiding their patients towards appropriate therapeutic avenues. While the treatment approaches for DMG in the USA, Europe, and Australia share similarities and prioritize a multidisciplinary approach, several key groups are driving significant advancements in DMG research and treatment. Notably, the United States hosts the largest network of centers specializing in DMG treatment, encompassing biopsy, molecular classification, and clinical trials. In addition, Spain and Switzerland in Europe, as well as Australia in the Oceania region, have centers that adhere to high-quality standards in DMG treatment. These regions are contributing substantially to the understanding and management of DMG. Collaborative efforts by prominent groups such as the Pediatric Brain Tumor Consortium (PBTC), the Children’s Oncology Group (COG) Phase I/Pilot Consortium, and the Pacific Pediatric Neuro-Oncology Consortium (PNOC) are actively involved in DMG research and treatment.

This review emphasizes the need for highly specific and tailored approaches to effectively target DMGs and improve patient outcomes. Molecular profiling and personalized medicine hold great potential for optimizing treatment strategies and improving prognosis for patients with DMGs.

## 5. Conclusions

Despite the recent advancements in the diagnosis and treatment of DMG, it continues to be a complex and challenging disease. However, the recent advances in the molecular classification of DMG have opened new avenues for research.

MRI remains the main diagnostic method. However, other techniques, such as biopsy and cerebrospinal fluid, plasma or serum analysis, are growing in importance due to the advent of molecular studies of new biomarkers that provide valuable prognostic insight.

Regarding therapy, the current standard of care, which combines radiation therapy and chemotherapy, provides limited improvement in the overall survival of these patients. Early studies suggest proton therapy as a safer modality, especially in cases that require re-irradiation, as well as novel local administration techniques of drugs and radiation or BBB permeability increase approaches, that prove promising in preclinical studies.

Moreover, novel molecular-based targeted therapies bring new opportunities noteworthy of consideration. These techniques offer a broad variety of possibilities, from targeted therapies against histone methylation abnormalities to immunotherapies that range between single antibodies or nanobodies to whole cells, like CAR T-cells or CAR-NK therapies, and even oncolytic vaccines.

All in all, although DMG is still a clinically challenging tumor, these new strategies offer hope for the future. Continued research and clinical trials are necessary to fully understand the safety, efficacy, and long-term effects of this approaches in order to advance its clinical application and to establish effective and safe treatment protocols.

## 6. Limitations of the Study

The limitations of the study include the restricted timeframe of 1 January 2012 to 30 June 2023, which may exclude relevant studies published before or after that period. Language bias is present as only English and Spanish articles available online were included missing other languages. The study relies on specific databases (PubMed, Cochrane Library, and SciELO through the LILACS), potentially overlooking relevant studies from other sources. The broad inclusion criteria may lead to varying levels of evidence and quality in the selected studies. Publication bias may be present due to the inclusion of online articles. The findings may not be generalizable to other brain tumors or populations. These limitations should be taken into account when interpreting the study’s conclusions. 

## Figures and Tables

**Figure 1 jcm-12-05261-f001:**
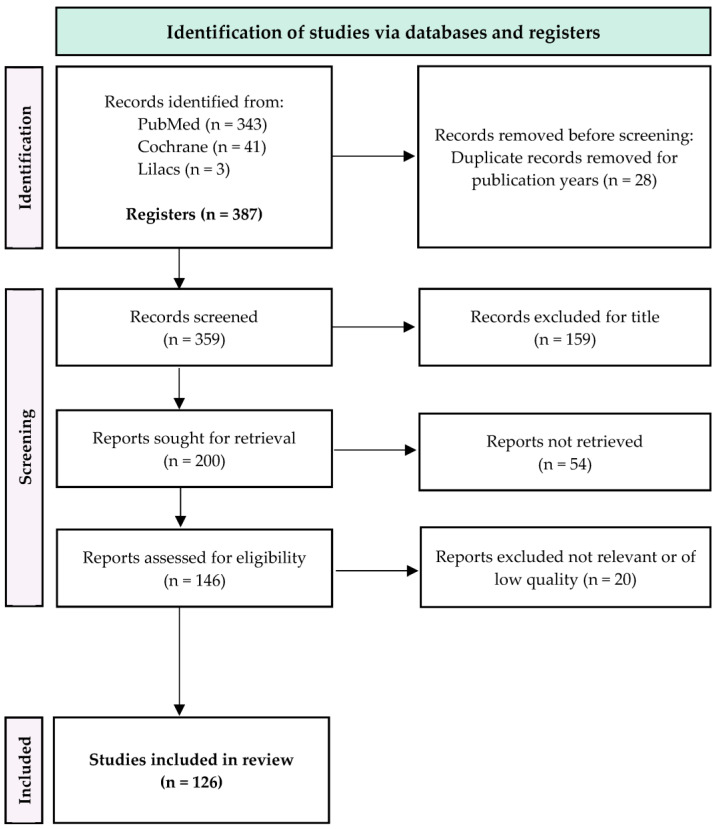
PRISMA flow diagram.

**Figure 2 jcm-12-05261-f002:**
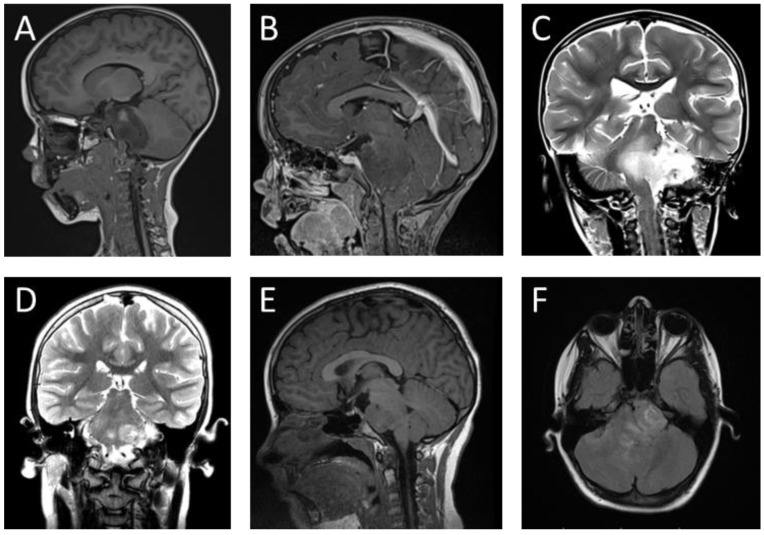
MRI imaging of DMG tumors. Patient 1: (**A**) DMG lesion on T1-weighted without contrast sagittal image, (**B**) DMG lesion on T1-weighted post-contrast sagittal image, (**C**) DMG lesion on T2-weighted coronal image. Patient 2: (**D**) DMG lesion on T2-weighted coronal image. (**E**) DMG lesion on T1-weighted without contrast sagittal image, (**F**) DMG lesion on FLAIR axial image.

**Figure 4 jcm-12-05261-f004:**
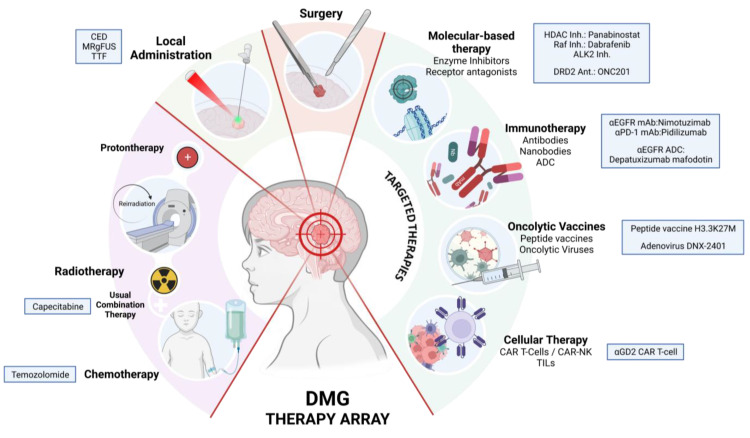
DMG treatment. Current and investigational therapies for diffuse midline glioma. The combination of chemoradiotherapy remains the standard of care with low overall survival rates.

**Table 1 jcm-12-05261-t001:** Frequent genetic modifications in DMG with histone 3 mutations.

Amplification	Mutation	Deletion
*PDGFRA* (30%) *CDK4/6* or *CCND1-3* (20%) *MYC/PVT1* (15%)	*TP53* (30%) *ACVR1* (30%) *PPM1D* (15%) *ATRX* (15%)	*CDKN2A/B* (<5%)

Source [5,6,25,46,53].

## Data Availability

Not applicable.

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
