# Peer review of "Epidemiology, Diagnostic Strategies, and Therapeutic Advances in Diffuse Midline Glioma"

_jcm, 2023, doi:10.3390/jcm12165261_

Round 1

Reviewer 1 Report

jcm-2531206: 

“Epidemiology, Diagnostic Strategies, and Therapeutic Advances in Diffuse Midline Glioma

Brief Summary:

This is a systematic literature review regarding the diagnosis and treatment of Diffuse Midline Glioma (DMG). This literature review was performed from 2012 to 2023. This is an important review because the diagnosis of this tumor has completely changed since the WHO CNS5  was published in 2021. The change in diagnosis came from tissue sampling of this tumor. A specific genetic mutation (H3K27) was identified that grouped different midline tumors with similar prognosis (WHO Grade 4). Thus, the name from Diffuse Intrinsic Pontine Glioma (DIPG) was changed to Diffuse Midline Glioma (DMG).

Specific Comments:

P. 1 lines 37-44 (Introduction):

  • More emphasis should be made in the introduction of the difference in diagnosis of DIPG and DMG. WHO Grade 4 tumors in the thalami, brainstem, and spinal cord of children and young adults are all DMG if they possess the mutation H3-K27. The addition of these tumors to DIPG’s makes the patient demographic different and it is now not only a pediatric disease, but also of young adults.

P. 3 lines 119-121 (Methods):

  • During the literature review, one search term used was H3, instead of H3-K27. H3-G34 is the mutation that is found in Diffuse Hemispheric Glioma. Thus, using H3 alone might make the search too broad.

P. 4 lines 138-140 (DMG Definition):

  • “Primary affects children between the ages 3 and 10.”

    • This is not the true definition of DMG. With the addition of other midline structures to DIPG, the diagnosis of DMG is more broad. Thus, one must take into the account the location of the tumor to accurately talk about demographics/epidemiology.

P. 6 lines 186-189 (Biopsy and laboratory diagnostic techniques)

  • Stereotactic biopsy has been available for many years, but only more recently has biopsies been performed for DMG. Is there any evidence in your literature review of how this has evolved? New approaches? New hardware/software? For example, obtaining tissue via the middle cerebellar peduncle approach.

P. 10 lines 318-322 (Chemotherapy)

  • In most centers in the US chemotherapy is not used for the treatment of DMG. However, it would be interesting to emphasize the differences between Europe and the US. Are there any differences in outcome with different algorithms of treatment? What are your center's current treatment algorithms? This information is important because it can provide valuable treatment options for this rare tumor.

P. 12 lines 417 to 432 (Tumor treating fields):

  • The example of implanting tumor treating fields in the clivus is very interesting. However, the main example of tumor treatment fields is placed on the scalp. Has this been studied for DMG? Do tumor treatment fields reach the midline structures if it is placed on the scalp?

P. 14 (Scope and future directions):

  • It is important to mention where clinical trials for DMG are being held. Certain patients are willing to travel for treatment. Thus, geographic location of trials and their differences may help physicians guide their patients in the right direction for treatment.

Reviewer 2 Report

This is a review on „Epidemiology, Diagnostic Strategies, and Therapeutic Ad-

vances in Diffuse Midline Glioma“. After a structured screen of the literature by the authors, 72 articles were included in the review.

I would suggest to restructure the abstract in a way that also emphasises the mist  important diagnostic (technical) and therapeutic developments. 

In the introduction, page 2 line 45, aphasia should be erased as a symptom; this is not a brain stem symptom.

The introduction can be shortened, as parts are redundant. E.g., the use of biopsy in the new diagnostic era is mentioned several times.

The chapter on biopsy (3.3.2) is superficial and could include some more information on neurosurgical techniques as well as the goal of not inducing harm by the biopsy.

in the section on molecular studies, (3.3.3), the diagnostic procedures is only touched. Common histological markers and the use of methylation (850K), mRNA expression etc. arrays are not even mentioned. The Heidelberg classifier has to be cited. Subtypes ah H3K27 that constitute the new WHO diagnosis are not mentioned. Instead, liquid biopsies are mentioned as a new way of diagnosis. Liquid biopsies may be promising, but are far away from standard clinical use. In addition, several therapeutic modalities are mixed into theischapter that should be mentioned later on in the therapeutic section. I am not sure why the use of nanobodies is discussed in that extent in the same chapter. This has nothing to do with diffuse midline glioma.

In chapter 3.4, authors pick up the new classification. In my view, there is no need to also show a table on the old classification. A short summary, however, could be helpful. Here, IMPACT-NOW could be mentioned as a driver of development for the new classification. Also, authors state that the non-H3K27 gliomas have a better prognosis and suggest that this has biological reasons; in my opinion, this is more due to the midlife localisation of H3K27 which induces severe symptoms and prevents effective treatments as resection.

In the chapter on radiotherapy (3.5.1), the authors spend almost half of the text volume on an experimental study that uses magnetic resonance spectroscopy to predict outcomes. This is far away from being clinically relevant, and as MRS methods fail for decades to have impact on brain tumor research, the message should be shortened. In addition, MRS is not a radiotherapy, but a diagnostic MRI method.

In the chapter on chemotherapy (3.5.3), the sentence that „Radiosensitizers such as gemcitabine and capecitabine are commonly used, although their impact on overall mortality is minimal or negligible“ is not connected to any citations. Please add some. In the same and other chapters, if authors cite positive results, it would be good to also see the number of patients and therapeutic effect (HR, p) behind that. 

The chapter on local administration of therapies (3.4.5) is longer than the chemotherapy chapter, despite the fact that local therapies have not improved outcomes yet and are specifically difficult to performed in midline gliomas. I therefore wonder if a more critical view on this would reflect the area of research in a more adequate way.

In the chapter on targeted therapies (and also on the diagnostic chapter), the prevalent co-occurence of H3K27 and BRAFV600 mutations, that constitutes a valid therapeutic target, is not mentioned. Please add. In the same chapter, a number of targets and therapeutic agents are mentioned that in part have no connection to diffuse midline glioma, e.g. IL13Ralpha2 and EGFR. These should be erased, and targets ant therapeutic modalities should be separated from each other. It is also equivocal to cite one bioinformatic study that attaches to cancer neuroscience, but not to discuss the approach of Ca-channel targeting in a  translational context. Do authors think that a treatment that targets the ubiquitous Ca channel will be a good way to treat tumors, specifically midline gliomas? If yes, they should provide strong arguments for their opinion. ONC-201 is an important new substance and is available within an early access program in the US. It should therefore be discussed in more detail. 

In the immunotherapies chapter (3.5.7), peptide vaccines that were in clinical trials are not mentioned. T- (NK-CARS are the most advanced means of immunotherapies and should placed at the end of the immunotherapy chapter. Oncolytic virus-based therapies often rely on peri-operative intratumoral injection of viruses. This limitation that is important for diffuse midline gliomas should be discussed. Immun checkpoint inhibitors should be discussed in a separate chapter.

Figure 4 is generic and has no specific view on midline gliomas. It should be adapted accordingly.

The list of references includes 81 citations, but the authors mention 72 publications they used for their review. Please explain.

In summary, It has to be restructured according to the suggestions above. In addition, text has to be weighed according to comprehensive and reliable criteria, e.g. relevance of an approach for current clinical use or scientific relevance / interest for the future. In addition, all information should be erased that is not connected to diffuse midline gliomas, and redundancies should be reduced.

None

Round 2

Reviewer 1 Report

Amazing work with your review and adding my suggestions. Thank you for your kind words and I hope to see more of your contributions into scientific literature.